# Prediction of Content Success and Cloud-Resource Management in Internet-of-Media-Things Environments

Yeon-Su Lee [1], Ye-Seul Lee [1], Hye-Rim Jang [1], Soo-Been Oh [1], Yong-Ik Yoon [1] and Tai-Won Um [2,*]

[1] Department of Bigdata Analysis & Convergence, Sookmyung Women's University, Seoul 04310, Korea; pluto2002p99@sookmyung.ac.kr (Y.-S.L.); lystanz@sookmyung.ac.kr (Y.-S.L.); hyerim@sookmyung.ac.kr (H.-R.J.); tnqls2306@sookmyung.ac.kr (S.-B.O.); yiyoon@sookmyung.ac.kr (Y.-I.Y.)
[2] Graduate School of Data Science, Chonnam National University, Gwangju 61186, Korea
* Correspondence: stwum@jnu.ac.kr

**Abstract:** In Internet-of-Media-Things (IoMT) environments, users can access and view high-quality Over-the-Top (OTT) media services anytime and anywhere. As the number of OTT platform users has increased, the original content offered by such OTT platforms has become very popular, further increasing the number of users. Therefore, effective resource-management technology is an essential aspect for reducing service-operation costs by minimizing unused resources while securing the resources necessary to provide media services in a timely manner when the user's resource-demand rates change rapidly. However, previous studies have investigated efficient cloud-resource allocation without considering the number of users after the release of popular content. This paper proposes a technology for predicting and allocating cloud resources in the form of a Long-Short-Term-Memory (LSTM)-based reinforcement-learning method that provides information for OTT service providers about whether users are willing to watch popular content using the Korean Bidirectional Encoder Representation from Transformer (KoBERT). Results of simulating the proposed technology verified that efficient resource allocation can be achieved by maintaining service quality while reducing cloud-resource waste depending on whether content popularity is disclosed.

**Keywords:** content popularity; KoBERT; Sentiment Analysis; reinforcement learning; OTT; cloud computing

## 1. Introduction

The global emergency related to COVID-19 has substantially changed people's lives [1]. Due to the global pandemic and sequential lockdown, there has been a substantial increase in the use of the Internet and other online services for watching high-quality content using streaming services while moving wirelessly [2].

The Internet of Things (IoT) as a technology has attracted attention in this context. The IoT is made possible through the installation of various types of wireless communication modules such as mobile communication, Wi-Fi wireless LAN (Local Area Network), and Bluetooth modules that connect wired and wireless communication within a network. Data generated from IoT can be processed according to the form of the service field and transmitted to the cloud. This allows for high-quality services to be accessed anywhere in high-speed mobile environments through automated real-time decision-making based on data and pre-emptive responses based on prediction [3].

As the Over-the-Top (OTT)-service-platform market continues to expand, OTT service platforms require appropriate cloud service environments to store media contents and provide these contents according to users' requests [4]. However, it is difficult, time consuming, and expensive to build a cloud system on the OTT service platform itself. Therefore, the most recent OTT service providers have built OTT service platforms by subscribing to a cloud-computing service and renting computing and network resources rather than

directly implementing their own cloud environments [5]. By renting cloud resources to build an OTT service environment, OTT service providers can focus solely on media-service provision, thus allowing them to provide differentiated and competitive media services and reduce the cost and effort they invest in server operation and management [6]. Further, OTT service platforms based on cloud services can offer increased user satisfaction by providing media services effectively and stably.

OTT service platforms constantly need to provide a wide range of new content. OTT service users pay close attention to the presence or lack of new OTT content. Therefore, an OTT service platform is expected to show a flexible use pattern depending on the popularity of the new content [7]. Since popular contents are accessed by many users in a short period of time, OTT service providers need to prepare more service resources than usual. This differentiated response method is an important factor that affects the quality of services [8]. If the OTT service platform fails to properly provide media services when there is a surge in service requests caused by popular content, then content playback will be cut off, which will significantly decrease user satisfaction. Therefore, OTT service platforms need to be prepared to control resources and stably provide high-quality videos according to the success of the new content at times during which users' demands substantially fluctuate.

The purpose of this study was to analyze media-content comments in terms of emotion in order to predict success. A model that could appropriately allocate cloud resources to OTT service platforms according to the level of predicted success was also proposed. If the success of OTT service content can be predicted in advance and additional resources can be allocated to prepare for the rush of users' content requests, then the service quality can be stably maintained, thus increasing user satisfaction.

The rest of this paper is organized as follows. Related works are provided in Section 2. The proposed OTT-service-platform architecture and the prediction algorithm of content success are described in Section 3. Section 4 investigates the performance of the proposed resource-allocation algorithm through deep reinforcement-learning-based simulations. Finally, the results are summarized in Section 5.

## 2. Related Work

### 2.1. Cloud-Resource Allocation Based on Reinforcement Learning

To achieve efficient cloud-resource allocation, various perspectives must be considered. In particular, the user's resource requirements for an OTT platform may vary depending on the type of device, network, content, and user's own characteristics [9–11].

Therefore, in this section, we will examine factors affecting resource requirements according to device and network types, OTT user characteristics, and content types. We will describe cloud-resource allocation based on reinforcement learning. Cloud computing is activated as platforms that use virtualization technology, such as OTT services, are expanded. Cloud computing is a service that allocates and shares VMs (virtual machines) such as hardware and software on demand, thus allowing efficient resource management by lending as many resources as necessary. Since cloud-resource allocation may vary depending on the device or network, optimal services should be provided in consideration of various factors including CPU, memory, disk, and communication bandwidth [12]. Thein et al. [13] have proposed a system that can efficiently allocate resources to users' needs in consideration of cloud resources (including CPU computational capabilities, network bandwidth, and memory) and meet the Service Level Arrangement (SLA) set by cloud users. Gazdar et al. [14] have proposed a proactive protocol called the New Optical Proactive Protocol (NOPP) based on the optimal schedule for video segments and minimal streaming bandwidth. To this point, research aiming to improve the service quality of OTT platforms has mainly been focused on managing network traffic and maximizing performance.

Meanwhile, to determine the most efficient resource allocation possible for minimum service provisioning, it is important to analyze not only the types of resources, but also the user's consumption behaviors [11]. However, resource-allocation studies considering the user's consumption behaviors are insufficient [15]. Elagine et al. [11] have regarded

it as 'good' if the content being consumed is played at a normal speed according to the user's playback speed and 'bad' if it is fast forwarded through at a higher playback speed. Rojas et al. [15] have introduced a model that guarantees stable network performance in consideration of characteristics such as behavior patterns among OTT service users over time and content download rates over time. Lee et al. [4] have shown that resource allocation through reinforcement learning is efficient by predicting a user's resource request according to the user's rating. However, existing studies have not considered the success of new content as a factor in resource allocation. Recently, Netflix's original content 'Squid Game' has gained widespread global popularity, which has caused users to flood Netflix with access requests [7]. Despite the rapid increase in resource demand caused by the attention given to the newly announced original content, extant studies have only considered user consumption profiling without considering the success of the new content. Therefore, we aim to predict the success or failure of new OTT content, predict users' resource needs, and accordingly propose a model of efficient resource allocation.

### 2.2. Prediction of Content Success through Sentiment Analysis of Social-Media Big Data

With the myriad of data generated in digital environments, texts generated by users on web pages are used as important data in marketing, management, academia, and industry [16,17]. Sentiment Analysis is a text-mining technique that extracts the emotions of the person who wrote the text in the form of unstructured data. This Sentiment Analysis makes it easy to judge reputation and popularity because it makes it possible to analyze emotions contained within the text [18]. Sentiment Analysis can therefore predict users' reactions. This is valuable information as positive or negative reactions to specific content are closely related to the future use of that content. Therefore, positive or negative reactions in comments on OTT content previews on YouTube can be used to predict the success or failure of a new OTT content, thus allowing users' resource needs for new OTT content to be predicted in advance. Hammou et al. [19] have proposed a real-time processing framework that can emotionally analyze social big data based on distributed cyclic neural networks. This framework proves that social-media Sentiment Analysis is generally related to customer engagement and business performance.

Kumar et al. [20] have automatically generated a description of the decision making proposed by machine learning using the human-center artistic intelligence (HAI) technique and identified business opportunities based on this description. Since this approach focuses on decision making based on background, it emphasizes that the basis for classification is more important than the identification of Sentiment Analysis.

In this study, users' reviews were subjected to Sentiment Analysis. Since the amount of access to the OTT platform is flexible, it is useful because it can respond immediately, and users' reviews that are written in real time can be applied to resource allocation by analyzing real-time data. Ivan et al. [21] have analyzed online sources and achieved high performance in real-time changes in bitcoin investment. In their work, a DSS (Decision Support System) was implemented by applying HAI. However, Sentiment Analysis was used in our study because it was determined that the method of classifying emotions appearing in words was more appropriate than DSS to analyze positive or negative responses of the content to be applied.

Sentiment Analysis often involves classifying emotions into positive and negative emotions based on machine learning [22]. Supervised learning is a technique of machine learning. Among the studies using supervised learning, Peruson et al. [23] and Pak et al. [24] have improved SNS data using a Multinomial Naive Bayes algorithm. However, recent studies using Sentiment Analysis based on deep neural networks have achieved higher performance than studies using Sentiment Analysis based on machine learning [25]. This is because deep learning is advantageous for guessing latent functions and deriving approximations based on various input data. The performance improves when there is an increase in the number of dimensions of requirements used as input to the Sentiment-Analysis classifier.

Specific examples of deep-learning techniques using BERT include models released by Google [26], which have respectively learned BookCorpus in 800 million words and Wikipedia corpus in 2.5 billion words through MLM (Masked Language Model) and NSP (Next Sentence Prediction). After the initial learning, the word is tokenized, the sentence is split into words, and segment embedding and position embedding are created by the element-wise sum [27,28]. As a form of Sentiment Analysis using KoBERT (Korean Bidirectional Encoder Representation from Transformer) [29], which is specialized in Korean, Eom et al. [30] have developed a comment classifier using KoBERT for online political-opinion analysis. Park et al. [31] have analyzed comments on recent YouTube videos using the KoBERT model. Choi et al. [32] have used KoBERT that was trained by adding Korean wiki and news to implement a Sentiment-Analysis model based on e-commerce data. KoBERT was created by the SK Telecom AI Research Team, T-Brain. It applies a data-based tokenization technique to reflect characteristics of irregular language changes in Korean, resulting in performance improvement of more than 2.6% while only using 27% of the tokens that were used by a previous one [33]. Despite recent research interest in OTT content, studies conducted to date have not dealt with content analysis of the OTT platform, which is a major limitation. It is also difficult to find a study that has applied analysis results to cloud-resource allocation. Therefore, in this study, YouTube comments (positive or negative) related to OTT content were analyzed using the KoBERT technique. Through this analysis, the popularity of OTT content was predicted and applied to cloud-resource allocation.

## 3. Proposed OTT Cloud-Resource-Allocation Algorithm

In this section, the authors present a structural diagram to elucidate the process by which reinforcement-learning-based cloud resources are allocated on the OTT platform, which includes elements of predicting the success of OTT content in the proposed model. First, clause 3.1 describes the KoBERT-based comment-positivity classifier for content-success prediction on the OTT platform. The clause provides DQN (Deep Q Learning)-algorithm-based cloud-resource-allocation techniques combined with the proposed model's LSTM (Long-Short-Term Memory) neural network.

### 3.1. KoBERT-Based Comment-Positivity Classification for Content-Success Prediction

The proposed KoBERT-based comment-positivity classification can predict the movie box-office success of a movie on the OTT service platform. Sentiment analysis using KoBERT largely proceeds through stages of (1) data collection, (2) data preparation, (3) model learning, and (4) model evaluation. The key to this approach is to 'learn' a method of classifying certain areas so that computer programs can perform tasks, such as distinguishing emotions (i.e., positivity), and then classify emotions for new data based on this learning. First, in the process of collecting and preparing data, the data to be learned are prepared. In the model-learning step, previously prepared data are input and parameters for appropriately classifying the data are found. Finally, in the model-evaluation stage, the performance of the model is evaluated based on the new data.

To classify positive and negative comments related to the prediction of the box-office success of OTT content, additional learning was performed by adding a linear neural network according to the success prediction of the content. To analyze positive and negative emotions in comments, a classifier was added to the pre-trained KoBERT model for further fine tuning, as shown in Figure 1.

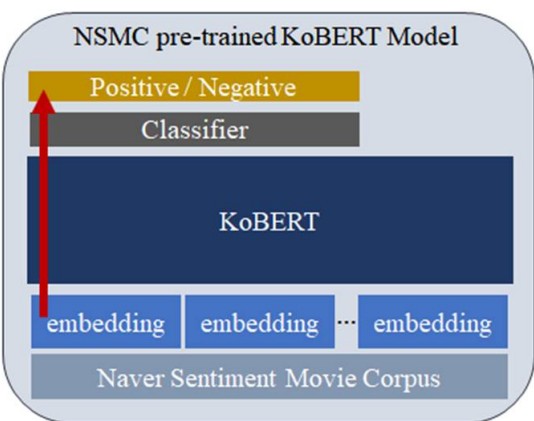

**Figure 1.** KoBERT-based Classification Architecture for Positive and Negative Comments.

First of all, in the data collection step, NSMC (Naver Sentiment Movie Corpus) [34] was collected as a fine-tuned dataset related to Korean movie-review comments. The information in the dataset consisted of comment ID, comment content, and emotion type (positive, negative). The comment ID is the ID of the movie-review author. The comment content consisted of up to 140 characters. The emotion type was scored as 0 for positive and 1 for negative. The classification of positive and negative is described as follows. With scores from 1 to 10, 1 to 4 were said to be negative while scores of 9 to 10 were positive. The remaining scores of 5 to 8 were judged as neutral comments and were thus excluded [31].

As described above, the dataset containing the correct answer value goes through two steps before the model-learning step. The first step is tokenization. The KoBERT model is a model that learns sentences and words from the Korean wiki and the Korean news in the neural-network structure of the BERT (Bidirectional Encoder Representation from Transformer) model. Therefore, the tokenization process is performed in the same way as BERT. The second step involves dividing the tokenized dataset into three datasets: learning, verification, and evaluation sets. NSMC data were used. These data consisted of 150,000 training data and 50,000 evaluation data. Datasets were divided into training datasets and evaluation datasets. Some evaluation datasets were classified as verification datasets for model verification during learning.

After collecting the data, the algorithm coefficient was updated through the learning dataset. The learning was properly performed through the verification dataset. For example, the accuracy and error alert of the training dataset were constantly improved. If the verification dataset does not show the same improvement, the model will determine itself to be over-fitted to the data and therefore stop learning from previous iterations [29].

In the data-learning stage, the linear neural network will be further advanced to enable the classification of positive and negative comments on a movie. In this process, parameters such as batch size, epoch size, learning rate, maximum length of tokens and number of categories were set. Since OTT comments in this study were classified as positive and negative, a binary classification was set.

Finally, the accuracy of the model was checked through the evaluation dataset. Since the performance of the model needs to be checked, the evaluation dataset is completely separated from learning using non-contaminated datasets.

Through the positive comment classifier learned by NSMC, a positive analysis was performed on YouTube comments related to the OTT content. Figure 2 shows a structural diagram of the YouTube comment positive–negative classifier related to the KoBERT-based OTT content that was learned by NSMC.

The sentiment classifier for YouTube comments collected YouTube comments related to the OTT content. Elements such as emoticons and punctuation marks were removed from the collected data through preprocessing, and comments written in foreign languages were excluded. After entering YouTube comments into the KoBERT model trained by NSMC, the user's positive response to the content was classified. As a result of the classification,

the contents with many positive reactions were classified as box-office successes and those with many negative reactions were classified as box-office failures.

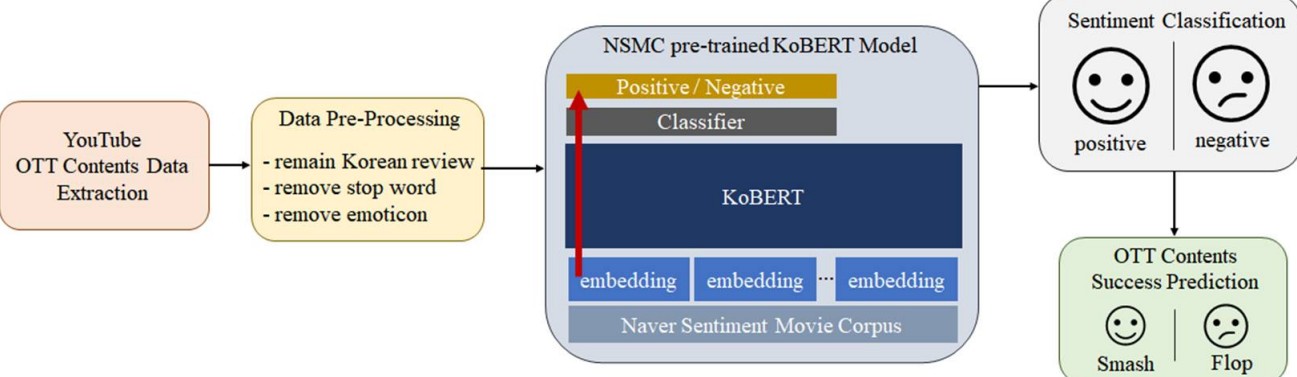

**Figure 2.** KoBERT-based YouTube Comments Sentiment-Analysis Prediction Architecture.

*3.2. Structural Diagram of Cloud-Resource Allocation Based on Reinforcement Learning*

　　This section describes the proposed model of a cloud-resource allocation in an IoMT-based OTT platform. The cloud-resource-allocation structure diagram of the proposed OTT platform is shown in Figure 3 below.

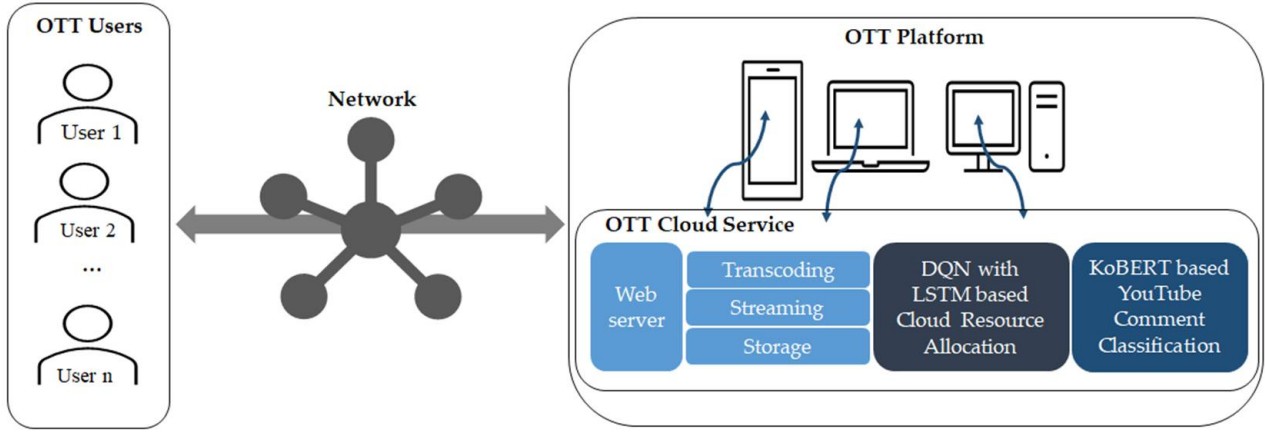

**Figure 3.** Cloud-Resource-Allocation Architecture in OTT Platform.

　　First, users request video services from the OTT platform through a network. The OTT platform requires resources according to the usage of OTT users from OTT cloud services. The use of OTT cloud services enables OTT users to use effective content by providing cloud-resource allocation based on reinforcement learning to the OTT platform.

　　At this time, the OTT cloud service consists of services such as a web server to provide cloud-resource-allocation services, transcoding to provide resources tailored to the environment of the OTT platform, storage to store the backup data of the OTT platform, streaming to provide real-time content on the OTT platform, and cloud-resource allocation based on reinforcement learning. Since this study aims to allocate cloud resources based on reinforcement learning in the OTT platform, a cloud-resource-allocation technique combined with LSTM and DQN was used. In the proposed technique, the DQN algorithm combining the LSTM neural network could predict resource usage by reflecting the OTT usage patterns of OTT users. Through this process, resources are effectively allocated to OTT users. The algorithms used to predict reinforcement-learning-based resource allocation on the OTT platform will be described in detail in Section 3.3.

### 3.3. DQN-Algoritm Model Combining LSTM Neural Network for OTT Platform Resource Allocation

OTT operators can predict resource usage by time using DQN-algorithm-based reinforcement learning which combines LSTM neural networks based on users' access patterns. When the predicted number of resources is requested from the cloud, the cloud provides the requested value to the OTT platform. The OTT platform then allocates the number of resources delivered to users. Figure 4 shows the structure diagram of the reinforcement-learning-based resource-allocation model reflecting the successful prediction of OTT content.

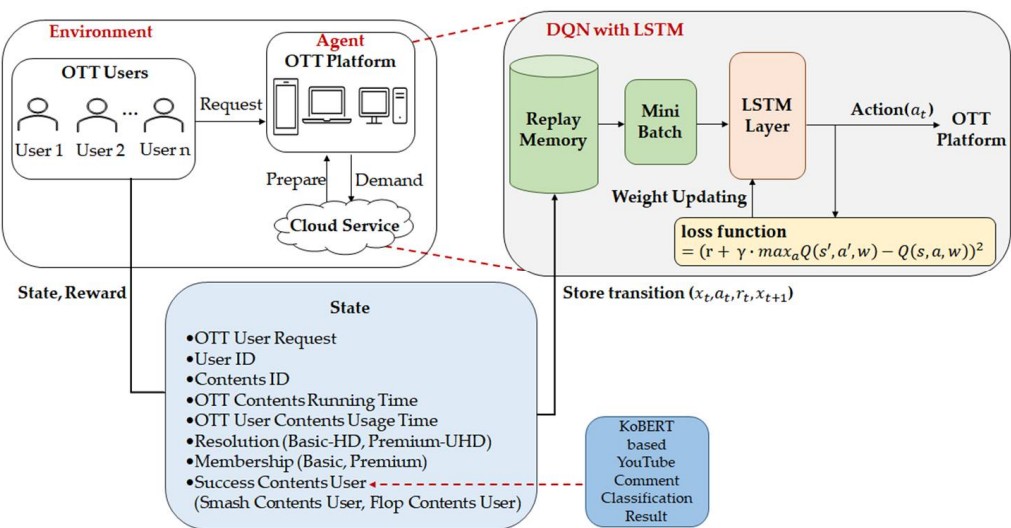

**Figure 4.** DQN-Based LSTM OTT Platform Cloud-Resource-Allocation Architecture.

Reinforcement learning is defined in terms of MDP (Markov Decision Process) elements, such as state (S), behavior (A), state-transition probability (P), compensation (R), and subtraction rate (Y). Since this study was related to the usage of the OTT platform, information that could be used to recognize the user's resource request, content success, and system usage over time was reflected in the state (S). The state (S) consists of Request, Contents Success User, User ID, Contents ID, Running Time, and Time. Request is the amount of resources required by users. Content Success Prediction is divided into Smash for box-office success and Flop for box-office failure according to the movie's box-office prediction. User ID and Contents ID are identification numbers of the user and OTT content, respectively. Running Time is the playback time of the OTT content. Finally, Time is the user's access time. Types of behavior (A) include ADD for adding resources from the cloud, REMOVE for removing unnecessary resources, and HOLD for maintaining current resources. Table 1 lists the definition of each element used in the proposed model.

**Table 1.** Definitions of Model Components.

| | Definition |
|---|---|
| | Possibility of all states |
| S | {Request, Membership (Basic, Premium), Success Contents User (Smash Contents User, Flop Contents User), Resolution (HD, UHD), User ID, Contents ID, Running Time, Time} |
| | Possibility of all actions |
| A | {ADD, REMOVE, HOLD} |
| P | Probability of moving to the next state when action a is taken in state s |
| R | A reward that only evaluates the current state and behavior |
| Y | The value of the reward awarded |

In this study, the agent aims to flexibly allocate cloud resources according to the workload of the OTT platform. As depicted in Figure 5, the DQN-based LSTM comprises three stages that are fundamental for optimizing the OTT resource-allocation system.

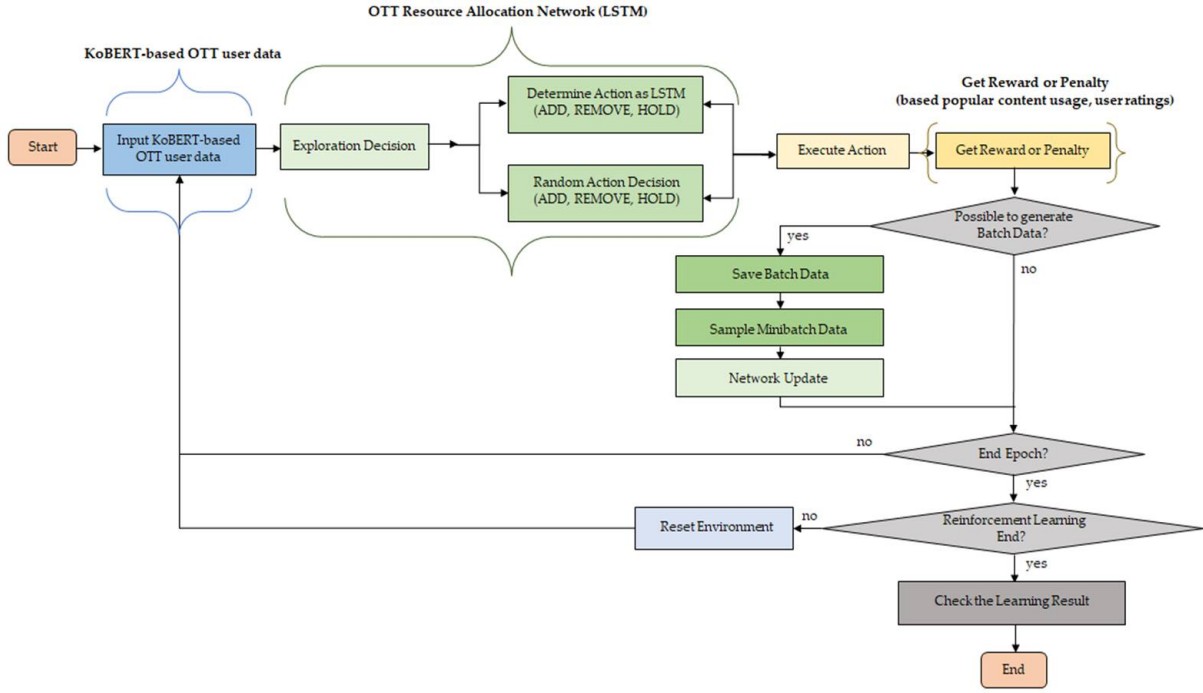

**Figure 5.** Flowchart of DQN-based LSTM OTT Platform Cloud-Resource-Allocation Structure.

The proposed DQN-based LSTM OTT platform cloud-resource-allocation structure is designed as follows. First, data reflecting the results of KoBERT's Sentiment Analysis of YouTube comments are combined with OTT user data to generate input data. Second, the DQN-based LSTM model is performed with the input data. The action is determined either by the LSTM network or randomly. Third, the DQN-based LSTM model receives a reward or penalty according to user rating and popular-content usage.

In reinforcement learning, agents are rewarded and punished according to their actions. They learn in the direction of maximizing rewards. $S_t$ is a set of values representing the current situation. S represents a set of all possible states. Behavior is an option we can take. The set of all possible actions is called $A(S_t)$. The selected action based on the state of $S_t$ at a specific time $t$ is called $a_t$. Rewards are benefits that follow when an agent takes an action. However, the reward is expressed in $r$ as an immediate value that only evaluates the current state and behavior. The behavior with the highest cumulative compensation should be selected to have the behavior with the best results in the long run.

$$Q_\pi(s,a) = E_\pi[r_{t+1} + \gamma Q_\pi(s_{t+1}, a_{t+1})|s_t = s, a_t = a] \tag{1}$$

The Bellman Equation (1), which is recursively expressed by defining the expected value of cumulative compensation as the behavioral value function of the corresponding state $S$, is as above:

$$Q_\pi^*(s,a) = max_\pi Q_\pi(s,a) \tag{2}$$

The optical policy is a policy with the largest expected value of cumulative compensation among several policies, as shown in Equation (2). In the next state $s'$, the $Q$ function is updated using the largest $Q$ value. Sufficient exploration is then performed while selecting the behavior with the highest $Q$ value when determining the behavior. Exploration uses a greedy technique that operates intensely at the beginning of learning but decreases toward the end of learning:

$$loss\ function = \left(r + \gamma max_{a'}Q(s', a', w) - Q(s, a, w)\right)^2 \tag{3}$$

However, if the state and behavior have high dimensions because they are expressed in the form of a *Q*-table, then an overlap problem may occur. To prevent this problem, a DQN algorithm that combines the LSTM neural network was used. The DQN algorithm expressed the state space through the network instead of the existing *Q*-table and applied the Experience-Replay technique to allow the state–action value function to converge. DQN aims to approximate the optical policy and learns in the direction of minimizing loss function (3) through stochastic gradient descent.

To store the data in the minibatch format, a replay memory is created and the *Q*-value is initialized. OTT user-related data are received as input data. Such data consist of user information, OTT content information viewed by the user, success of OTT content, and required resource amount items. Next, the probability of the behavior output from the LSTM neural network is used to select the behavior (whether to ADD, REMOVE, or HOLD) and determine the number of resources to change. Based on the determined behavior, the OTT platform then compares the number of predicted resources held with the number of requested resources to determine whether provisioning is successful and to give compensation. Penalties are given when under-providing or over-providing occurs. Rewards are given when appropriate provisioning is made. Meanwhile, resources are provided differently according to the user's rating subject to resource provision. Lastly, when using Smash content, greater rewards are given regardless of the rating so that resources are provided differently according to the degree of content success that the user is using.

The overall process of the algorithm proposed in this study is presented in Algorithm 1.

---

**Algorithm 1: Reinforcement Learning Algorithm (LSTM & DQN)**

---

Initialize replay memory D to capacity N
Initialize action-vale function Q with random weights
For episode = 1, M do
For $t$ = 1, T do
Apply LSTM classifier
input $x$ is the KoBERT-based OTT data
calculate forget gate $f_t$, input gate $i_t$, intermediate cell state $\widetilde{C}_t$
update cell state $C_t$
calculate output gate $o_t$
output is the action probability $\epsilon$
With the probability $\epsilon$, select an action $a_t$ and decide resource units
if $a_t$ is ADD, OTT platform adds resource units
else if $a_t$ is REMOVE, OTT platform removes resource units
else if $a_t$ is HOLD, OTT platform keeps current resources
Execute action $a_t$ and observe $r_t$
Set $s_{t+1} = s_t$
Store transition $(x_t, a_t, r_t, x_{t+1})$ in replay memory D
Sample random minibatch of transition $(x_t, a_t, r_t, x_{t+1})$ from replay memory D
Set $y_j = \begin{cases} r_j & \text{if episode terminates at step } j+1 \\ r_j + \gamma max_{a'}Q\left(x_{j+1}, a'; \theta\right) & \text{otherwise} \end{cases}$
Compare the actual resource request and the predicted resource request
If the provisioning is optimal, $r_j$ is positive, otherwise $r_j$ is negative
Perform a stochastic gradient descent
End for
End for

---

## 4. Performance Analysis

This study categorized popular content using a KoBERT-based YouTube comment classifier. Based on the above classification, reinforcement learning from the OTT provider's perspective was used to analyze the performance of predicting the amount of cloud-resource requests and the efficient allocation of resources. Section 4.1 describes the results of the performance analysis for YouTube comment classifiers based on KoBERT. Section 4.2 describes the results of cloud-resource-allocation-performance analysis using reinforcement learning that reflects whether viewers will watch popular content.

### 4.1. KoBERT-Based YouTube Comments Classifier

NSMC conducted a positive analysis of YouTube comments related to OTT content through the learned positive comment classifier. Table 2 lists the parameters and accuracy set by the proposed KoBERT-based YouTube comment classifier.

**Table 2.** KoBERT-based YouTube Comment Parameters.

| Type | Parameters |
|------|------------|
| Batch Size | 64 |
| Epoch Size | 5 |
| Learning Rate | $5 \times 10^{-5}$ |
| Maximum Length | 64 |
| Number of Category | 2 |

Table 3 presents results of learning of each model using NSMC, including LSTM, the BERT-multilingual model, and the KoBERT model. In the case of LSTM, after converting each syllable of NSMC data into a vector, a padding operation was performed to equally match the sentence length. When learning by stacking the LSTM layer in one layer, it showed a performance accuracy of about 85%. The BERT-multilingual model is a type of BERT model. It learned dictionaries in 104 languages. The BERT-multilingual model showed a performance accuracy of about 87%. On the other hand, KoBERT is characterized by learning 5 million additional Korean wiki sentences and 20 million Korean news sentences in the BERT-multilingual model. It understands characteristics of the Korean language better than the BERT-multilingual model. The KoBERT model showed a performance accuracy of about 89%, which was about four percentage points higher that the LSTM, and about two percentage points higher than the BERT-multilingual model.

**Table 3.** Model Accuracy.

| Model | Accuracy |
|-------|----------|
| LSTM | 85% |
| BERT-multilingual | 87% |
| KoBERT | 89% |

Based on model-accuracy results, the performance of KoBERT was confirmed to be outstanding. Figure 6 shows an example of KoBERT's positive and negative responses to YouTube comments. Comments are scored as 1 for positive reactions and 0 for negative reactions. Since KoBERT conducts Sentiment Analysis for Korean, examples of comment analysis show the results of Sentiment Analysis for Korean comments.

| | |
|---|---|
| 이렇게 재있는 드라마 오랜만이다 | 1 |
| 서진씨의 엄청나고 웅장한 연기력과 그렇지 못한 수련씨의 귀여움 | 1 |
| 민설아 범인은 진짜 누구인지 궁금함 | 0 |
| 하 주석훈 로나 공주님안기 하는거 설레서 숨이 제대로 안쉬어짐 후후 | 1 |
| 코미딘가 재있는데 왜케 웃기지 특히 마지막 | 1 |
| 아 주단태 심수련 왜 자꾸 설레지 | 1 |
| 드라마 잘 안보는데 이건 진짜 안불수가 없다 | 0 |
| 솔직히 도비서랑 서진이 잘어울리심 | 1 |
| 진지한데 웃기다 | 1 |
| 전이랑 뭐가 다른가 했는데 다크써클이었군요 안가린게 훨씬 예쁘시네요 완전 국보급 외모 | 1 |
| 김소연 눈연기 대박 소오름 | 1 |
| 아닐 심수련은 그대로 심수련인데 강마리는 왜 끔겄억 강마는 누구입니까 근데 강마리 유동필 보면서 무슨 생각할까 참 참담하겠다 | 0 |
| 심폐소생술도 안끝내고 뽀뽀하는거 나만 놀랬남 | 0 |
| 누가 진천댁을 하대해 아 진짜 이런 발성 처음이야 너무종아 | 1 |
| 와 회장님 딕션진짜 | 1 |
| 석경이 시즌 이 더 예쁘지만 언제나 예쁜 한지현 배우님 | 1 |
| 와 이 때 록련 얼굴합 미쳤다 로건 하얀셔츠 너무 잘 어울러 | 1 |

**Figure 6.** Example of KoBERT-based YouTube Comments Results.

Table 4 shows the analysis of positive and negative classifications of YouTube comments on content based on the learned KoBERT.

**Table 4.** KoBERT-based YouTube Comment-Classifier Results.

| Movie | Positivity | Negativity |
|---|---|---|
| Race of Freedom: Um Bok Dong | 40.59% | 59.41% |
| Parasite | 60.63% | 39.37% |
| Penthouse2 | 43.04% | 56.96% |
| The Silent Sea | 67.85% | 32.15% |

As a result of comment analysis, 40.59% of comments for 'Race of Freedom: Um Bok Dong' were classified as positive and 59.41% were classified as negative. Considering 'Parasite', 60.63% of comments were classified as positive and 39.37% as negative. In the case of 'Race of Freedom: Um Bok Dong', the box office failed with a cumulative audience of about 170,000 [35]. In the case of 'Parasite', the analysis was well performed in that it achieved a cumulative audience of over 10 million [36].

Meanwhile, in the case of the drama 'Penthouse 2', it was confirmed that it was well classified as a popular content because it was successful with a viewer rating of 20% or more [37]. On the other hand, Netflix's original series 'The Silent Sea' was classified as fraudulent content which was highly popular. However, based on the IMDb tally, it was confirmed that 16% of the audience gave a score of 1 to 3 points. Some foreign media gave it failing scores [38].

### 4.2. KoBERT-Based DQN Alogithm Model Combining LSTM Neural Network for OTT Platform Resource Allocation

In the case of the proposed cloud-resource-allocation technique, an OTT service-platform simulator based on reinforcement learning was implemented by modifying and expanding the open-source program [39] related to deep reinforcement learning. The developed simulator consisted of an agent, data manager, environment, policy learner, policy network, and main module.

The agent module is an agent class for performing resource-allocation-related actions and managing the number of resources. The environment module manages the amount of resource requests over time. The module receives information on resource requests and returns popular-content usage and ratings used by users from OTT data. The policy learner module has a class that implements a reinforcement-learning method. The main module executes resource-allocation reinforcement learning.

The agent module was modified and implemented to suit the proposed model. The simulator was executed by changing the Mbps in the agent module. The number of virtual machines and the resulting cloud cost were measured and analyzed.

The OTT data used in the simulator were generated by reflecting the user's OTT service-usage pattern and KoBERT's content-positive response for 24 h in the actual OTT service. Based on the actual case of a surge in the number of simultaneous users when

popular content is released [40], it is assumed that 60% of all users request newly released box-office content, thus reflecting KoBERT's positive response. After conducting KoBERT, about 60% of users responded positively to popular content. This was reflected in the OTT user-data composition. Data variables included access time, user identifier, movie identifier, user departure, user resource request, user's resource return, accumulated resource demand, user rating, and popular-content use.

The operation process of the simulator is described as follows. The virtual user requires resources from the OTT platform. At this time, the user's rating is set differently based on whether it is Basic or Premium. Mbps is allocated differently in the cloud service according to the use of popular content and the user's rating. The number of resources required by the user varies with time. The time zone is set from 0 to 24 h in the unit of 1 h. The number mentioned above is defined as Request. Request differentially reflects Mbps according to the rating of OTT users and their usage of popular content with the number of resources requested by OTT users. The request made by the virtual user is then transferred to the agent OTT platform. The agent receives information from the corresponding environment again, determines the behavior, and receives information on compensation. For learning, a DQN algorithm that combines the LSTM neural network is used. The OTT platform determines behavior through the learned reinforcement of the learning algorithm. Actions include ADD, REMOVE, and HOLD.

VM for resource allocation was defined using Mbps units. The use of Mbps units as the basis for resource allocation was based on Optimizing Auto-Scaling Virtual Machines for a Cloud-based VoD (Video on Demand) Data Center [41] and on VM types such as Amazon EC2 M6g [42]. Table 5 shows the server and research environment in which the simulator for performance evaluation was executed.

**Table 5.** Server Environment.

| Type | Environment |
| --- | --- |
| Operating System | Windows 11 |
| CPU | Intel® Core™ i7-8750H CPU @ 2.20 GHz |
| RAM | 16384 MB RAM |
| GPU | NVIDIA GeForce GTX 1060 |
| Library | Tensorflow 2.6.0 |
| | Keras 2.6.0 |
| Programming Language | Python 3.9.7 |

Figure 7 shows performance-evaluation results by VM type reflecting only the user's rating and whether to use popular content based on KoBERT. First, m5a instances with a maximum Mbps of 2880 per VM were designed. In the case of m5a.xlarge, as shown in the graph, except for 1 a.m., resource allocation appeared in a similar form to demand VM at 1 a.m., when the user's resource request increased more rapidly than the model that reflected only the user's rating. In addition, it was confirmed that the model reflecting KoBERT after 21 p.m. reflected the user's resource request more accurately than the model reflecting only the user's rating. After 20 p.m., the number of user's resource requests increased rapidly. Thus, the model reflecting KoBERT in this section was more accurate, meaning that the OTT user's resource-request prediction was better than the model reflecting only the user's rating.

Figure 8 shows the case of M6g.xlarge with a maximum Mbps of 4750 per VM, the KoBERT reflected model, except for 1 a.m., VM can be secured in preparation for user resource requests expected to increase rapidly after 20 p.m., indicating that resource-allocation prediction is better than user ratings alone. In the case of 8 a.m. to 12 p.m., there is a difference between the KoBERT model and the model that reflects only the user's rating. At this time, in the case of the model that reflects only the user's rating, resource prediction cannot be considered efficient in that resource allocation increases in the area where the user's resource request decreases.

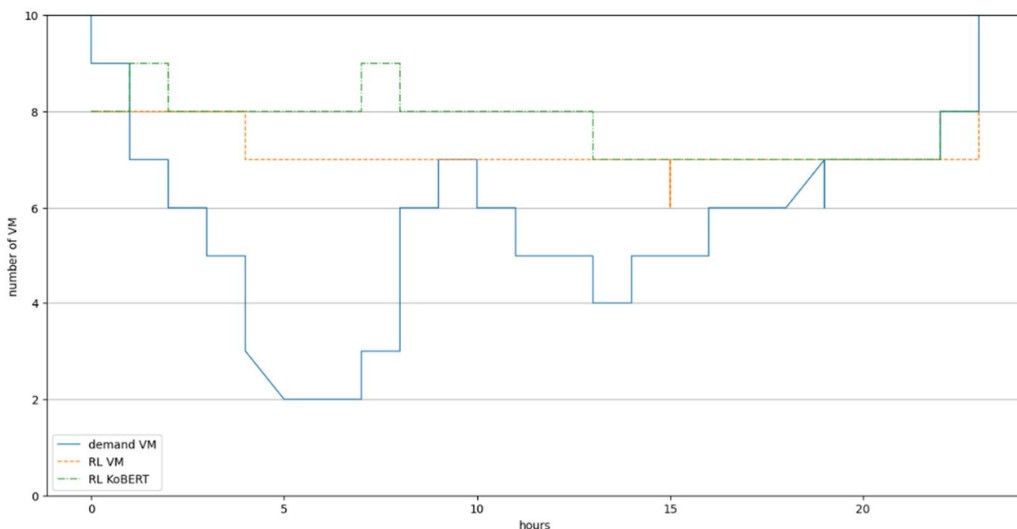

**Figure 7.** Resource Allocation VM Comparison According to User's Rating and Resource Requirement Reflecting the Use of Popular Content (Maximum Mbps 2880 per VM).

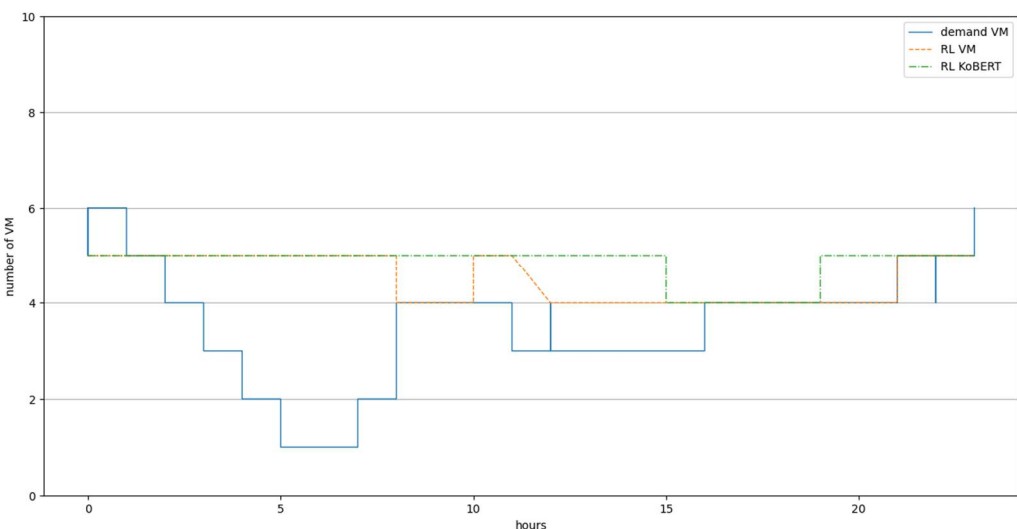

**Figure 8.** Resource Allocation VM Comparison According to User's Rating and Resource Requirement Reflecting the Use of Popular Content (Maximum Mbps 4750 per VM).

Figure 9 shows the case of M6g. 8xlarge with up to 9000 Mbps per VM, the model reflecting KoBERT maintains the VM at 3 and allocates resources according to the user's resource request. The model reflecting only the user's rating was dynamic in that it changed from 15 p.m. to 18 p.m. compared to the model reflecting KoBERT.

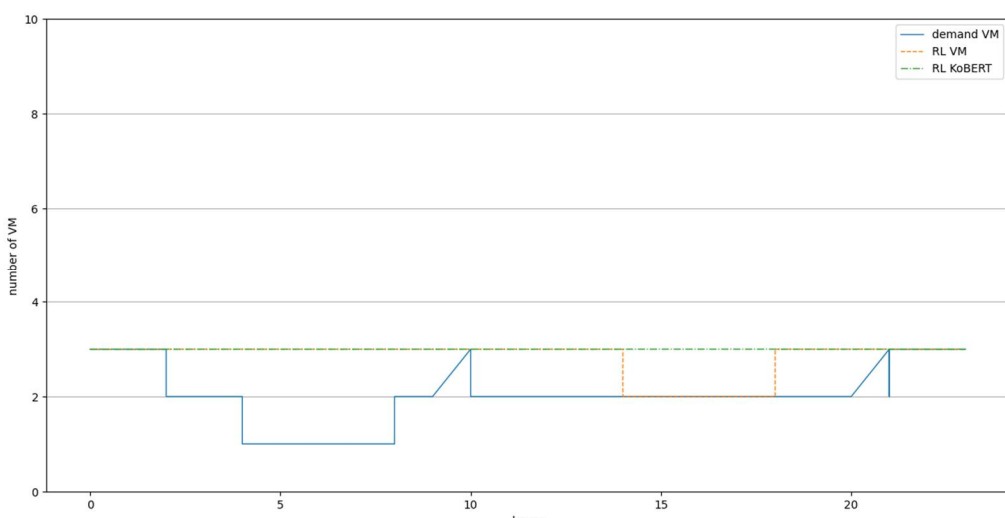

**Figure 9.** Resource Allocation VM Comparison According to Resource Requirements Reflecting User's Rating and in terms of Whether to Use Popular Content (Maximum Mbps 9000 per VM).

The graph in Figure 10 shows results of examining resource allocation reflecting the user's rating and the use of popular content according to resource requirements for each VM type. The situation in which the maximum support Mbps was 4750 was found to be more reliable VM resource allocation for resource requirements than other values of Mbps such as 2880 and 9000. Furthermore, from a cost point of view, it is most efficient when Mbps is 4750.

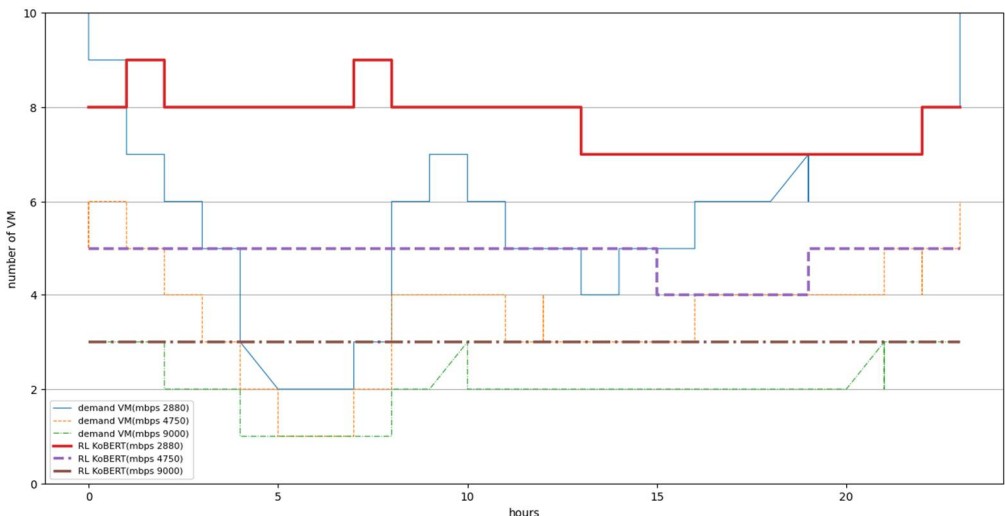

**Figure 10.** VM Resource Allocation According to Mbps Reflecting Resource Requirement, User's Rating, and Whether to Use Popular Content.

Figure 11 shows results of calculating the cloud usage cost per hour of each VM type algorithm according to the resource requirement reflecting the actual Amazon ec2's rate policy of each VM type. M6g.xlarge can support up to 4750 Mbps. In terms of cost efficiency, when M6g.xlarge was used, it was confirmed that resources were efficiently allocated at 33.54% for 2880 Mbps and 48.87% for 9000 Mbps.

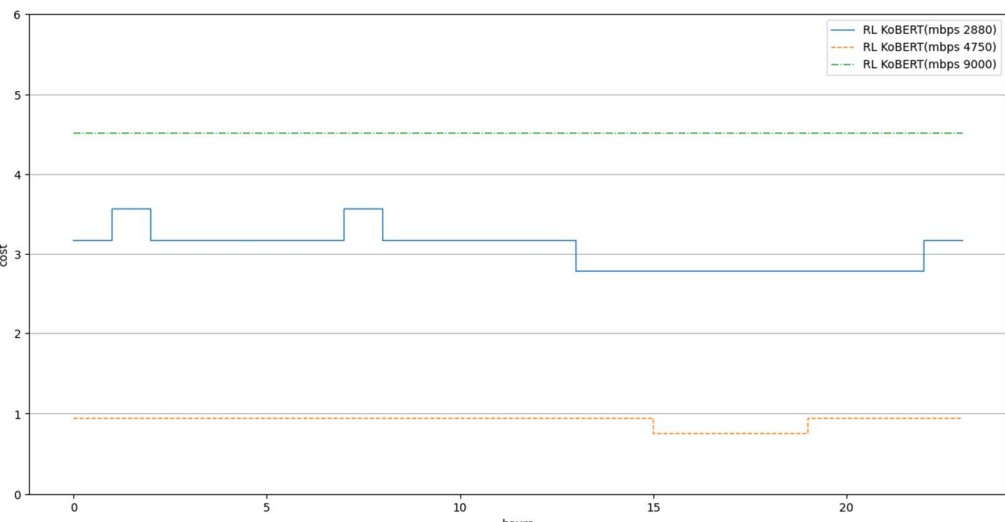

**Figure 11.** Cloud Costs for Each VM Type Reflecting Resource Requirements, User's Rating, and Use of Popular Content.

## 5. Conclusions

This study designed a model that can allocate appropriate cloud resources to the OTT platform by predicting the user's demand for OTT popular content. The proposed model is a resource-allocation technique that can accurately reflect characteristics of OTT, reduce OTT operators' expenses, and ensure the quality of media services provided on the platform. Performance evaluation of the proposed model showed that when using the model proposed in this research, the cloud-usage costs per hour compared according to VM type and resource were efficiently allocated at 33.54% for Mbps 2880 and 48.87% for Mbps 9000.

Several directions for future research are suggested below. First, it is necessary to verify the performance by reflecting various aspects, such as the amount of time needed to provide OTT services to users and the playback time. Secondly, it is necessary to verify performance by diversifying types of popular content tested. Finally, if individual users' tastes and behavior patterns are reflected in the model, then the pre-prediction accuracy can be improved, thus allowing resource-allocation problems to be solved more efficiently.

**Author Contributions:** Conceptualization, Y.-S.L. (Yeon-Su Lee); Formal analysis, Y.-S.L. (Yeon-Su Lee), Y.-S.L. (Ye-Seul Lee), H.-R.J. and S.-B.O.; Funding acquisition, T.-W.U.; Investigation, Y.-S.L. (Ye-Seul Lee); Methodology, Y.-S.L. (Yeon-Su Lee) and Y.-S.L. (Ye-Seul Lee); Resources, Y.-S.L. (Yeon-Su Lee) and Y.-S.L. (Ye-Seul Lee); Software, Y.-S.L. (Yeon-Su Lee), H.-R.J. and S.-B.O.; Visualization, S.-B.O. and H.-R.J.; Supervision, T.-W.U. and Y.-I.Y.; Validation, T.-W.U.; Writing—original draft, Y.-S.L. (Yeon-Su Lee) and Y.-S.L. (Ye-Seul Lee); Writing—review & editing, T.-W.U. and Y.-I.Y. All authors have read and agreed to the published version of the manuscript.

**Funding:** This work was supported by the National Research Foundation of Korea (NRF) grant funded by the Korea government (MSIT)(NRF-2018R1A2B2003774) and Institute of Information & communications Technology Planning & Evaluation (IITP) grant funded by the Korea government (MSIT) (No. 2021-0-02068, Artificial Intelligence Innovation Hub).

**Conflicts of Interest:** The authors declare no conflict of interest.

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
