# Peer review of "Prediction of Content Success and Cloud-Resource Management in Internet-of-Media-Things Environments"

_electronics, doi:10.3390/electronics11081284_

Round 1

Reviewer 1 Report

Article Title: Content Success Prediction and Cloud Resource Management in Internet of Media Things Environments

This paper proposes a technique for predicting and allocating cloud resources as an LSTM-based reinforcement learning method that reflects whether to watch popular content using KoBERT from the perspective of OTT service providers. However, I have the following concerns regarding this manuscript:

  • The abstract is failed to present the detailed information of the standard abstract formatting: (e.g., introduction/context, aim/objectives, methodology, results, conclusion, and future work).
  • The introduction section is too short and its' need to be more precise. It is not clear exactly to the reader what is the knowledge gap, problem or puzzle motivates the research and this manuscript. Where exactly is the gap that this paper will fill? what is the research question? what is the contribution of this paper? and what exactly the solutions reported in the existing literature couldn't offer? The proposed problems are sketchy and unclear.

Author Response

Point 1: I The abstract is failed to present the detailed information of the standard abstract formatting: (e.g., introduction/context, aim/objectives, methodology, results, conclusion, and future work).

Response 1: According to the above comments, it was revised as follows.

Abstract:(Introduction)In the IoMT (Internet of Media Things) environment, users can access and view high-quality OTT (Over-The-Top) media services anytime and anywhere. As the number of OTT platform users has increased, the original content offered by such OTT platforms has become very popular, thus further increasing the number of users. Therefore, effective resource management technology is an essential aspect of reducing service operation costs by minimizing unused resources while securing the resources necessary to provide media services in a timely manner when user’s demand resource ratings change rapidly. However, previous investigated, efficient cloud resource allocation without considering the number of users after the release of popular content. (Aim)This paper proposes a technology for predicting and allocating cloud resources in the form of an LSTM (Long Short-Term Memory)-based reinforcement learning method that provides information to OTT service providers about whether users are willing to watch popular content using KoBERT (Korean Bidirectional Encoder Representation from Transformer). (Result)The results of simulating the proposed mode verified that efficient resource allocation can be achieve by maintaining service quality while reducing cloud resource waste depending on whether or not content popularity is disclosed.

Point 2: The introduction section is too short and its' need to be more precise. It is not clear exactly to the reader what is the knowledge gap, problem or puzzle motivates the research and this manuscript. Where exactly is the gap that this paper will fill? what is the research question? what is the contribution of this paper? and what exactly the solutions reported in the existing literature couldn't offer? The proposed problems are sketchy and unclear.

Response 2: According to the above explanation following description is added to the manuscript.

The global emergency related to COVID-19 has substantially changed people's lives [1]. Due to the global pandemic and sequential lockdown, there has been an substantial increase in the use of the Internet and other online services, such as for watching high-quality content through streaming services while wirelessly moving [2].

Accordingly, IoT (Internet of Things) is a technology that has attracted attention in this context. IoT is made possible through the installation of various types of wireless communication modules such as mobile communication, Wi-Fi wireless LAN, and Bluetooth modules that connect wired and wireless communication within a network. Data generated from IoT is processed according to the form of the service field and transmitted to the cloud. This allows for high-quality services to be accessed anywhere, in high-speed mobile environments through automated real-time decision-making based on data and preemptive response based on prediction [3].

As the OTT (Over-The-Top) service platform market continues to expand, OTT service platforms will require appropriate cloud service environments to store media contents and provide them according to users’ requests [4]. However, it is difficult, time-consuming, and expensive to build a cloud system on the OTT service platform itself. Therefore, most recent OTT service providers have built OTT service platforms by subscribing to a cloud computing service and renting computing and network resources rather than directly implementing their own cloud environments [5]. By renting cloud resources to build an OTT service environment, OTT service providers can focus solely on media service provision, thus allowing them to provide differentiated and competitive media services and reduce the cost and effort they invest in server operation and management [6]. Further, OTT service platforms that are based on cloud services can offer increased user satisfaction by providing media services effectively and stably.

OTT service platforms constantly need to provide a wide range of new contents, and OTT service users pay close attention to the presence or lack of new OTT content. Therefore, it is expected that an OTT service platform will show flexible use pattern depending on the popularity of the new content [7]. Since popular content is accessed by many users in a short period of time, OTT service providers need to prepare more service resources than usual, and this differentiated response method is an important factor that affects the quality of services [8]. If the OTT service platform fails to properly provide media services when there is surge in service requests caused by popular of content, content playback will be cut off, which will lead to significantly decreased user satisfaction. Therefore, OTT service platforms need to be prepared to control resources and provide high-quality videos stably according to the success of new content at times during which users' demands fluctuate substantially.

The purpose of this study is to analyze media content comments in terms of emotional to predict success, and to propose a model that appropriately allocates cloud resources to OTT service platforms according to the level of success predicted. If the success of OTT service content can be predicted in advance and additional resources can be allocated to prepare for the rush of users' content requests, then the service quality can be stably maintained, and user satisfaction can be increased.

The rest of this paper is organized as follows. Related works are provided in section 2. The proposed OTT service platform architecture and the prediction algorithm of content success are described in section 3. Section 4 investigates the performance of the proposed resource allocation algorithm through deep reinforcement learning-based simulations. Finally, the results are summarized in section 5.

Reviewer 2 Report

The authors propose a prediction-based strategy for cloud resource management. However, there are still many parts that can be improved.

  1.  The keywords QoS and SLA appear under the abstract. However, in the main text section, there is no further explanatory note on QoS and SLA, such as how it is defined and how well it works experimentally.
  2.  In the abstract, it is mentioned that "provide media services in a timely manner in a situation". However, I cannot see any metrics of delay in the experiment.
  3. In the introduction, the proposed method is described in short:"This paper proposes a model that appropriately allocates cloud resources to the OTT service platform based on the prediction of the success level of content through social media big data analysis." I cannot see any novelty from this short paragraph.
  4. The authors do not compare their proposed methods with others for both the prediction and resource management.
  5. In section 4, the different curves within the images are not well differentiated in the non-color print.
  6. How the virtual machine resources are related to the prediction results is not clarified, e.g., there is no description of how to manage the resources after the prediction results are obtained.
  7. The authors need to rewrite the following sentence:"At present, researchers usually employ three main methods in sentiment analysis, including the sentiment dictionary method, machine learning, and deep learning[18]. However, in actual use, creating and updating a sentiment dictionary is a difficult task, and the effect of a sentiment dictionary method relies on the quality of the sentiment dictionary itself [18]". Because it is exactly the same as other work.

Author Response

Thank you.

Reviewer 3 Report

This paper proposes a technique for predicting and allocating cloud resources using a reinforcement learning method that reflects whether to watch popular content from service providers. The manuscript still has some points which need revision.

  1. The abstract should briefly state the purpose of the research, the principal results and major conclusions. The abstract needs improvement in terms of clearly highlighting the contribution. It also needs to define the abbreviation and it should be grammatically revised.
  2. Some Keywords are unnecessary and they need to be shortened. Also, abbreviations are not needed to be defined in the keyword section.
  3. Abbreviations are repeatedly being defined. I recommend it to be revised where abbreviation should be defined only in the first instance mentioned starting from the introduction section.
  4. The background section can be further improved by giving appropriate and up-to-date references such as,
    • A machine-learning scraping tool for data fusion in the analysis of sentiments about pandemics for supporting business decisions with human-centric AI explanations SA Kumar, MM Nasralla, I García-Magariño, H Kumar PeerJ Computer Science 7, e713
    • A Repository of Method Fragments for Agent-Oriented Development of Learning-Based Edge Computing Systems I García-Magariño, MM Nasralla, J Lloret IEEE Network 35 (1), 156-162    3          2021
    • Real-time analysis of online sources for supporting business intelligence illustrated with bitcoin investments and iot smart-meter sensors in smart cities I García-Magariño, MM Nasralla, S Nazir Electronics 9 (7), 1101
    • A comprehensive analysis of healthcare big data management, analytics and scientific programming S Nazir, S Khan, HU Khan, S Ali, I Garcia-Magarino, RB Atan, M Nawaz IEEE Access 8, 95714-95733 34        2020
    • Healthcare Big Data Management and Analytics in Scientific Programming S Nazir, I García-Magariño, RB Atan, S Ali Scientific Programming 2021
  5. There are many grammatical and typo mistakes. The article should be carefully revised. 
  6. Some equation symbols need to be explained and defined in the text after each equation.
  7. It would be great to see some actual experiments of the conducted test by providing screenshots. 
  8. A flowchart of the proposed algorithm along with its description is needed
  9. The proposed method should be compared against other related works, and the authors should clearly identify how their proposed method is different from the others.

Author Response

Thank you.

Round 2

Reviewer 1 Report

I have read the revised version of the manuscript and I consider that the authors addressed all constructive recommendations from the previous round of review.

Author Response

Thank you for comment. Please see the attachment.

Reviewer 2 Report

The authors have revised the papers according to the comments. I suggest the authors highlight the advantages of their proposed method compared with others. In addition, they have to polish the paper further.

Author Response

(The authors gave the same response as above.)

Reviewer 3 Report

The comments were mostly addressed and I noticed the following works were discussed in the paper but not cited: 

"A machine-learning scraping tool for data fusion in the analysis of sentiments about pandemics for supporting business decisions with human-centric AI explanations" and

"Real-time analysis of online sources for supporting business intelligence illustrated with bitcoin investments and IoT smart-meter sensors in smart cities"

Please include them properly. 

Author Response

(The authors gave the same response as above.)

Round 3

Reviewer 3 Report

The authors addressed the comments.